# Evaluation of Control Effect of Confined Water Hazard in Taiyuan Formation Coal Seam Mining in Huanghebei Coalfield

**Jingying Li** [1,†], **Qingguo Xu** [2,†], **Yanbo Hu** [3,*,†] and **Xinmin Chen** [3]

1   Geo-Engineering Investigation Institute of Jiangsu Province, Nanjing 211102, China; ljyjsgk@163.com
2   Shandong Qiuji Coal Mine Co., Ltd., Qihe County, Dezhou 251113, China; xqg1970@163.com
3   School of Transportation Engineering, Nanjing Tech University, Nanjing 211816, China; xmchen@njtech.edu.cn
*   Correspondence: hyb@njtech.edu.cn
†   These authors contributed equally to this work.

**Abstract:** The shallow-layer resources in the Huanghebei Coalfield have been depleted, and the deep coal seam of the Taiyuan Formation (lower coal group) is the main continuous resource in mining at this stage. With the annual increase in mining depth, the exploitation of lower coal groups is being met with the influence of high ground stress, high water pressure, high temperature, and strong mining disturbances, as a result of which the threat of water inrush is particularly serious. Based on the grouting data of the coalface floor aquifer, this paper proposes an evaluation method for the control of water rushing into the coalface floor. By comparing the test data with mine electrical exploration data through ArcGIS, the results show that the water pressure threshold based on ArcGIS is twice the water pressure of the grouting reconstruction layer as the optimal solution. The research results can provide a reference for the prevention and control of water inrush in the lower coal group of Huanghebei Coalfield.

**Keywords:** water damage evaluation; water inrush; GIS; deep mining

## 1. Introduction

In the past 40 years of reform and opening-up, China's economy has developed rapidly, and coal resources have provided a great source of power. At this stage, the shallow coal resources in the North China Coalfield have been exhausted, and the deep mining mode is generally adopted. The mining of coal seams is affected by factors such as high ground stress, high water pressure, high temperature, and strong mining disturbance, among which the threat of high-pressure water hazards is particularly serious. In recent years, high-pressure water hazard accidents have occurred frequently, posing a huge threat to people's lives and property [1].

So far, scholars at home and abroad have made great achievements in the research of coal seam mining water hazard prevention and control. Among them, in the evaluation and early warning of water disasters in coal mining seam floors, V.F. Bens and H.E. Vanden Berg, etc., analyzed and studied the deformation mechanism of the fault zone in the surrounding rock and water spatial distribution using digital imaging [2]. Russian scholars have analyzed the fracturing of surrounding rocks in the process of mine water disasters, and proposed that the permeability of the surrounding rock was changed in the process of mine water disasters, ultimately leading to water disasters in adjacent coalfaces or adjacent mines [3]. American scholars established a structural unit model in 2009, which includes the core of the fault, the destruction zone and the original rock [4]. In 1989, Chinese researchers used information composite technology to comprehensively process multivariate geophysical information, and combined this with a geographic information system (GIS) to fit the factors affecting water floor bursting in coal seam mining, and established a water burst model for the mining area [5,6]. The water risk prediction of

the unmined area was carried out, and significant results were obtained. In 2007, Chinese scholars applied the weight determination method to the floor water burst factors, and combined this with a geographic information system to develop a fragility index method for the prediction and early warning of floor water bursting in coal seam mining [7–9]. In 2019, Chinese scholars devised the AHP-EW (Analytic Hierarchy Processing—Entropy Weight) combination to determine the weight of each factor influencing deep coal seam mining floor water bursting, and established a deep coal seam mining floor water burst risk evaluation model [10–12].

In terms of predicting the damage depth of the coal seam mining floor, Li et al. proposed the existence of the "lower three zones", i.e., the mining damage zone, the intact rock zone and the water pressure rising zone, causing coal mining floor damage [13–16]. Qian et al. proposed the key stratum theory for a coal seam mining floor water bursting mechanism, which strengthened the role of hard rock layers in suppressing water bursting, and pointed out that the layer of rock with the strongest bearing capacity above the water-bearing layer of the mining floor is the "key stratum" determining whether the mining floor bursts [17]. Miao et al. further explored Qian's "key stratum" theory and proposed the "water-separation key stratum" theory, which holds that the water-separation capacity of the coal seam floor does not only depend on the layer of rock with the strongest bearing capacity, but also relates to the combined action of the upper and lower rock layers connected with the key stratum [18–20]. At the beginning of the 21st century, Shi et al. studied the mechanism of mining floor water bursting and proposed that there are four zones under the mining floor: the mining pressure damage zone, the increased damage zone, the original damage zone and the original water rising zone [21–24]. They calculated the thickness of each zone using damage mechanics and fracture mechanics. Liu et al. combined the Coulomb–Mohr strength theory, the semi-infinite body theory, the Griffith strength theory, and a "two zone" model of mining floor water-conducting fracture zone and mining floor water-separation zone to obtain the maximum damage depth of the coal seam floor [25–29]. Meanwhile, they simplified the water-separation layer into a thin plate with uniform load around the four sides and obtained a prediction formula for the maximum water pressure that the mining roof can bear, using elastic–plastic theory. Chinese scholars first proposed the "in situ tension cracking" theory of coal seam floor damage following field tests of coal seam mining, and this states that in situ rock mass tension cracking from bottom to top forms in the pre-mining compressive stress section in the front of the mining field [30].

In terms of coal seam water control, due to the complexity of the coal-bearing geological conditions in China, on the basis of research on the mechanism of coal seam floor water inrush and safety evaluations and warnings, scholars and engineers in the field of engineering geology have also explored various types of mining water control methods. Up to now, China's main methods for coal seam floor water control have been "hydrophobic depressurization mining under pressure", "curtain interception", "grouting transformation", etc. In recent years, with the gradual increase in mining depth, most mines have gradually transitioned from "hydrophobic depressurization mining under pressure" to "grouting transformation" for controlling coal seam bottom water hazards. Chinese scientists conducted research on fault water disasters, comprehensively analyzed the fault water conductivity and regional water-richness of pressurized water-bearing strata, and adopted the methods of the grouting reinforcement of the fracture zone, the grouting transformation of the water-bearing strata on the upper and lower sides of the fault, and the dewatering and depressurization of the water-bearing strata to eliminate the occurrence of bottom water hazards [31,32]. Chinese scholars studied the grouting control of water-rich strata in weathering granite, simulating grouting reinforcement of the water-rich strata in weathering granite under different water pressure conditions, and the results show that the compressive strength and water stability of the specimens in the compaction zone increased with the increase in dewatering and depressurization [33–36]. When the dewatering and depressurization were 30%, the branch network slurry veins acted as

the compaction and support skeleton to improve the mine pressure strength and water stability characteristics of the compaction zone specimens. The Zhao Jiazhai Coal Mine 12,201 coalface quickly contained hazards related to the expansion of water by adopting a comprehensive scheme of high-pressure drilling grouting on the ground, supplementing grouting in the well and dewatering and depressurization, and effectively controlling the water outburst point [37–39].

As mentioned above, with the increasing depth of coal exploitation in China's central and eastern regions, systematic reports on the evaluation of the effect of water control in deep mining floor are yet to be seen.

This study mainly focuses on the F Coal Mine in the North of the Huanghebei Coalfield; the 1101 coalface reformed by directional drilling and grouting on the ground is taken as the research object. Based on the GIS system, water hazard control parameters, field survey and test information data, an evaluation model for water hazard control was established to analyze the water barrier effect of the coal seam bottom after grouting transformation. The results of this study can provide a reference for the prevention and control of confined water hazards in lower coal group mining in the Huanghebei coalfield.

## 2. Methodology

### 2.1. Overview of the Research Area

The F Coal Mine of the Huanghebei Coalfield is located in Qihe County, Dezhou City and Dong'e County, Liaocheng City, and belongs to Luxi Mining Group Co., Ltd. (Figure 1). The length of the mine is 5.7 km north–south and 7.2 km east–west on average, covering an area of 36.14 km$^2$ with approved mining elevations of +30 m to −900 m.

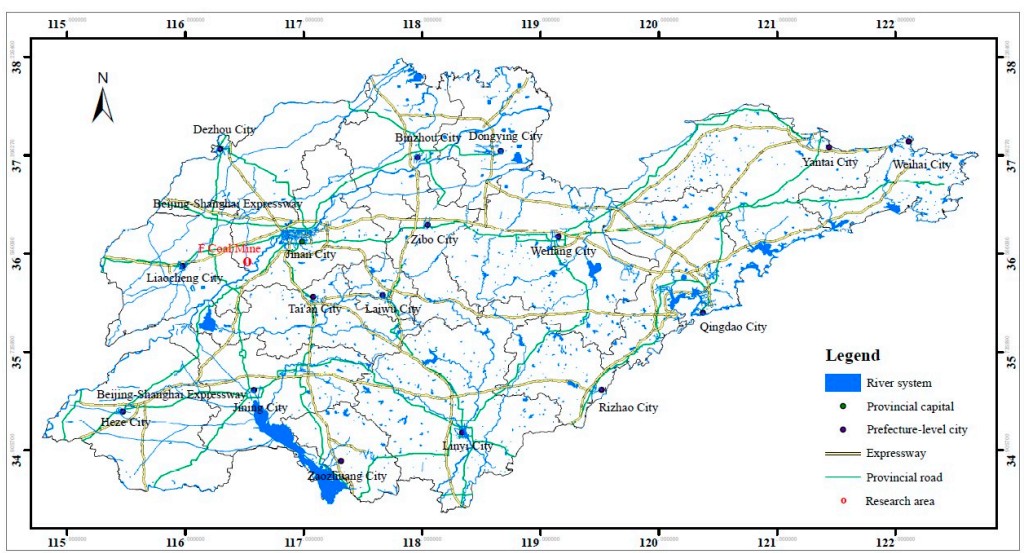

**Figure 1.** Map of the study area.

The production capacity of the mine is 450,000 t/a. There are 7#, 10#, 11# (NO. 11) and 13# (NO. 13) coal seams available for mining and partially available for mining in the mine. By the end of 2017, the 7# and 10# coal seams had been mined out, while the 11# coal seam is threatened by the top 4~5# limestone aquifers, floor Xujiazhuang limestone and Ordovician limestone karst aquifer, which pose a significant safety risk (Figure 2). In order to ensure the normal operation of the mine and reduce the economic risks of the project, it has been decided to carry out advanced regional governance of the 11# coal seam water hazards in the western mining area.

| System | Columnar legend | Thickness (m) | Lithology |
|--------|-----------------|---------------|-----------|
| Carboniferous | | 8.59 | Fourth and fifth layers of limestone |
| | | 1.80 | NO.11 coam seam |
| | | 7.44 | Siltstone |
| | | 4.00 | NO.13 coam seam |
| | | 5.19 | Mudstone |
| | | 19.58 | Siltstone |
| | | 8.60 | Xujiazhuang-limestone |
| | | 9.50 | Aluminous argillites |
| Ordovician | | 700.00 | Ordovician limestone |

**Figure 2.** Stratigraphic chart of the study area.

As shown in Figure 2, the distance between the No. 11 coal seam and the Xujiazhuang limestone aquifer is relatively small. Generally, the Xujiazhuang limestone aquifer and the Ordovician limestone aquifer can be regarded as the same aquifer (interconnected), with a high water pressure of 5–10 MPa. Under the disturbance conditions seen in the 11 coal mine, damage is easily caused to the coal seam floor, and this generates cracks. Under the combined action of mining disturbance and high water pressure in the limestone aquifer, it is easy for the coal seam floor rock to form a water channel. The Ordovician limestone aquifer can quickly inundate the entire mine (the Ordovician limestone aquifer is generally considered a strong water-rich aquifer, with a thickness of up to 700 m, which is the main threat to deep mining in North China's coal fields). Therefore, before mining the 11th coal seam, it is necessary to reinforce the bottom waterproof layer with grouting.

To date, the directional grouting transformation of the 1101 coalface floor has been completed (if there is a danger of water rushing in from the confined aquifer in the floor of the coal mining face, grouting can be generally carried out on the floor of the coal mining face before mining to increase the thickness of the water-resisting layer and ensure safe production). This research mainly focuses on the effect of the 1101 coalface floor water hazard remediation, establishing a scientific and effective evaluation model and verifying the accuracy of the evaluation model. The elevation range of the 1101 first mining face is $-339 \sim -422$ m, the width is 100 m, the length is 924 m, the coalface area is $9.24 \times 10^4$ m$^2$, the average coal seam thickness is 2.11 m and the average coal seam dip angle is 5° (Figure 3).

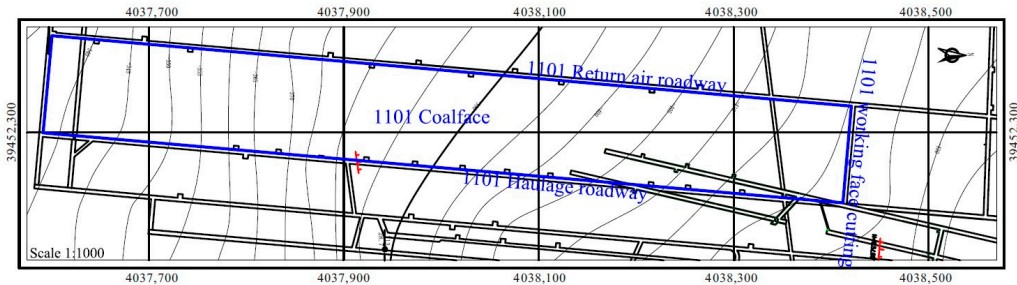

**Figure 3.** Layout plan of 1101 coalface.

### 2.2. Establishment of Evaluation Scheme

The principle of evaluating the grouting effect using GIS-based grouting pressure classification method is as follows.

Firstly, up to now, the main manifestation of the threat of water damage to the floor of deep coal seam mining in North China's coal fields is the Ordovician limestone aquifer. The deep coal seams in North China are generally located in the Taiyuan Formation of the Carboniferous System, and the Taiyuan Formation coal seams are relatively close to the Ordovician limestone aquifer. The Ordovician limestone aquifer is the coal-bearing basement of the North China coalfield, and is a high-pressure and strong water-rich aquifer, with a water pressure generally around 5–10 MPa. The disturbance caused by deep coal seam mining can lead to the destruction of the water barrier between the coal seam and the limestone aquifer, forming a water channel. The Ordovician limestone water will rush directly into the coal seam mining face through the water diversion channel, ultimately forming water inrush and causing disasters.

Secondly, the current most common prevention and control measures are to reinforce the bottom waterproof layer via grouting or to transform the top of the Ordovician limestone before mining the working face, thereby increasing the thickness and strength of the coal seam's bottom waterproof layer. The grouting effect determines the prevention and control of water damage on the bottom plate of the coal mining face.

Finally, the effectiveness of grouting is directly related to the grouting pressure. If the grouting pressure is too small, it is not possible to fully seal the cracks and dissolution gaps in the floor rock layer. If the grouting pressure is too high, it will expand the original rock fracture or cause the complete rock layer to rupture, forming a new water channel. Therefore, the magnitude of grouting pressure directly affects the effectiveness of grouting transformation. As such, the scientific classification and evaluation results of grouting pressure can be used to effectively evaluate the degree of water hazard risk.

A large amount of the data from deep mining in North China's coalfields indicates that the grouting sealing effect is optimal when the grouting pressure is generally set at around 2 to 2.5 times the water pressure. This research model is based on a large number of practical cases, and the grouting pressure of the three schemes is set at around 2.5 times the water pressure. Based on three schemes of class research, the classification results of different cases are compared and analyzed with the measured values, and the optimal scheme for evaluating the grouting effect can thus be obtained.

We assume that the grouting effect of each grouting point is determined by the risk evaluation of water rushing in from the floor of the coal mining face, as shown in Equation (1).

$$V = f(x, y) \tag{1}$$

where $V$ is the risk index of water damage to the bottom of the coal seam; $f(x, y)$ is the grouting pressure function evaluated by the GIS system; $x, y$ are geographic coordinates.

According to the principle of directional drilling grouting, after drilling is performed in the horizontal section, grouting is carried out, and the grouting interval is 200 m. According to experience with field construction, the grouting pressure has a strong correlation with the grouting effect. Therefore, the grouting pressure can be used to evaluate the grouting effect

at this time. The grouting and reforming data of the floor (Xujiazhuang limestone aquifer) of the 1101 coalface are shown in Table 1. The distance between Xujiazhuang limestone and the 11# coal seam is 34.73 m, and the Xujiazhuang limestone's elevation is about −415.23 m. According to past grouting experience, the water pressure of the Xujiazhuang limestone aquifer is 4.25 MPa, and the final pressure of grouting is generally 1.5 to 2.5 times that of the water pressure used for evaluating the grouting effect.

**Table 1.** Sample table of grouting dataset.

| Coordinate Data | | Grouting Pressure (MPa) | Coordinate Data | | Grouting Pressure (MPa) |
|---|---|---|---|---|---|
| X452437.5230 | 4038813.3490 | 12.5 | X452736.4278 | 4038528.3160 | 10.7 |
| X452486.9740 | 4038814.8070 | 11.5 | X452465.9240 | 4038513.8370 | 9.7 |
| X452551.5840 | 4038815.7270 | 10.5 | X452647.1340 | 4038501.4170 | 10.7 |
| X452616.2240 | 4038791.7870 | 11.0 | X452476.5040 | 4038472.6170 | 10.3 |
| X452691.3240 | 4038799.8670 | 10.5 | X452527.1340 | 4038468.6470 | 11.5 |
| X452422.3840 | 4038744.1370 | 10.0 | X452397.3540 | 4038434.5770 | 9.9 |
| X452514.7740 | 4038739.6570 | 10.0 | X452389.1240 | 4038426.3170 | 11.5 |
| X452558.8490 | 4038712.1650 | 10.5 | X452358.2940 | 4038413.2270 | 9.1 |
| X452632.3040 | 4038732.3570 | 10.0 | X452350.4540 | 4038392.6070 | 10.1 |
| X452729.1340 | 4038741.6870 | 11.5 | X452202.4180 | 4038675.7640 | 9.5 |
| X452632.3040 | 4038732.3570 | 10.0 | X452095.3780 | 4038457.3940 | 9.5 |
| X452729.1340 | 4038741.6870 | 11.5 | X452058.1480 | 4038379.9240 | 9.5 |
| X452661.3340 | 4038683.6270 | 11.0 | X452278.5880 | 4038706.4540 | 10.0 |
| X452727.4340 | 4038671.6870 | 10.1 | X452219.5380 | 4038617.8240 | 10.0 |
| X452401.7900 | 4038636.4498 | 10.0 | X452151.2580 | 4038517.0040 | 10.5 |
| X452420.8621 | 4038623.9074 | 10.0 | X452318.8980 | 4038671.3240 | 10.0 |
| X452461.6040 | 4038631.0770 | 10.5 | X452229.5380 | 4038562.5940 | 9.5 |
| X452547.8199 | 4038634.9909 | 9.5 | X452400.9434 | 4038658.9898 | 10.5 |
| X452539.4188 | 4038625.4052 | 10.0 | X452388.7580 | 4038647.8040 | 9.0 |
| X452791.7240 | 4038629.0870 | 10.6 | X452235.1183 | 4038509.2048 | 6.5 |
| X452482.5490 | 4038602.6720 | 10.0 | X452384.0080 | 4038563.4244 | 10.0 |
| X452565.7240 | 4038604.2770 | 10.0 | X452229.9085 | 4038458.0033 | 9.0 |
| X452640.9240 | 4038600.8270 | 9.8 | X452069.7166 | 4038346.5497 | 8.0 |
| X452382.5300 | 4038564.0498 | 10.0 | X452354.7337 | 4038469.1402 | 11.0 |
| X452645.2240 | 4038560.6670 | 10.0 | X452282.6517 | 4038435.1176 | 11.0 |
| X452416.0640 | 4038543.3770 | 10.5 | X452187.1680 | 4038389.8940 | 10.0 |
| X452472.7772 | 4038549.2587 | 7.0 | X452091.1480 | 4038345.8140 | 11.0 |
| X452539.6940 | 4038540.3670 | 10.0 | X452446.1880 | 4038427.0440 | 10.0 |
| X452584.7640 | 4038529.0570 | 10.4 | X452347.5180 | 4038399.4840 | 9.5 |

... ...

This study sets up three schemes for discussion, with the final grouting pressure set to 1.5, 2 and 2.5 times that of water pressure, respectively, for each experiment. Finally, the results of the geophysical exploration of the mine at the 1101 coalface are compared and analyzed to determine the evaluation criteria of the optimal final pressure.

Scheme I sets the final grouting pressure to 1.5 times that of water pressure, and the grouting effect of the Xujiazhuang limestone aquifer is divided into three levels according to the pressure: standard area (final pressure ≥ 7.875 MPa), qualified area (6.375 MPa ≤ final pressure < 7.875 MPa) and relatively weak area (final pressure < 6.375 MPa).

Scheme II sets the final grouting pressure to 2 times that of water pressure, and the grouting effect of the Xujiazhuang limestone aquifer is divided into three levels according to the pressure value: standard area (final pressure ≥ 10 MPa), qualified area (8.5 MPa ≤ final pressure < 10 MPa) and relatively weak area (final pressure < 8.5 MPa).

Scheme III sets the final grouting pressure to 2.5 times that of water pressure, and the grouting effect of the Xujiazhuang limestone aquifer is divided into three levels ac-

cording to the pressure value: standard area (final pressure $\geq$ 12.125 MPa), qualified area (10.625 MPa $\leq$ final pressure < 12.125 MPa) and relatively weak area (final pressure < 10.625 MPa).

## 3. Evaluation of Grouting Reconstruction Effect

### 3.1. Classification Result Based on GIS

The data in Table 1 are the original data from on-site grouting. We have used GIS to generate images according to the coordinates of each grouting point and the corresponding grouting pressure value (shown in Table 1), and then classified them according to the three experimental schemes set up, so as to get different results (as shown in Figures 4–6).

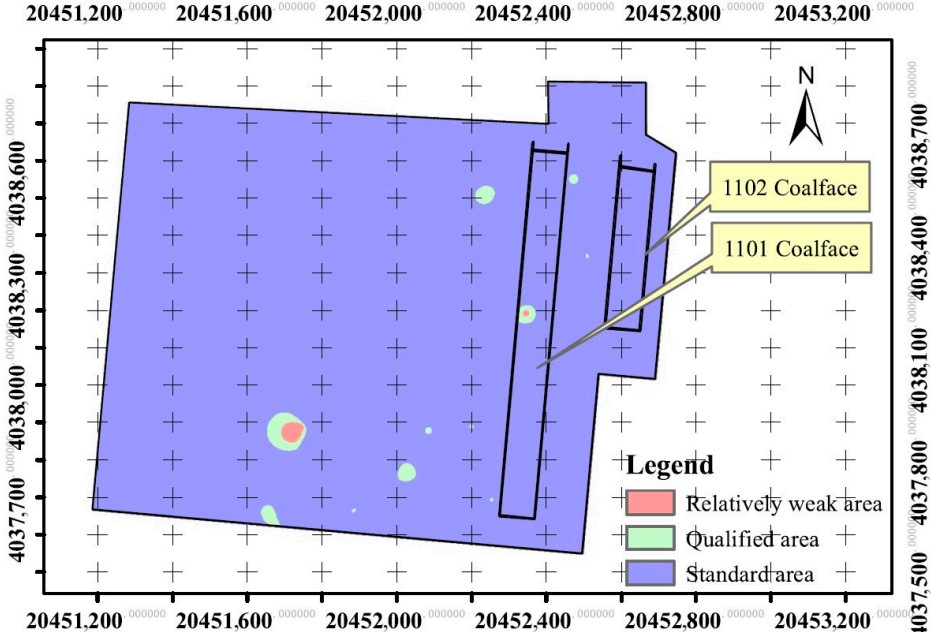

**Figure 4.** Evaluation diagram of grouting effect of 1101 coalface floor in scheme I.

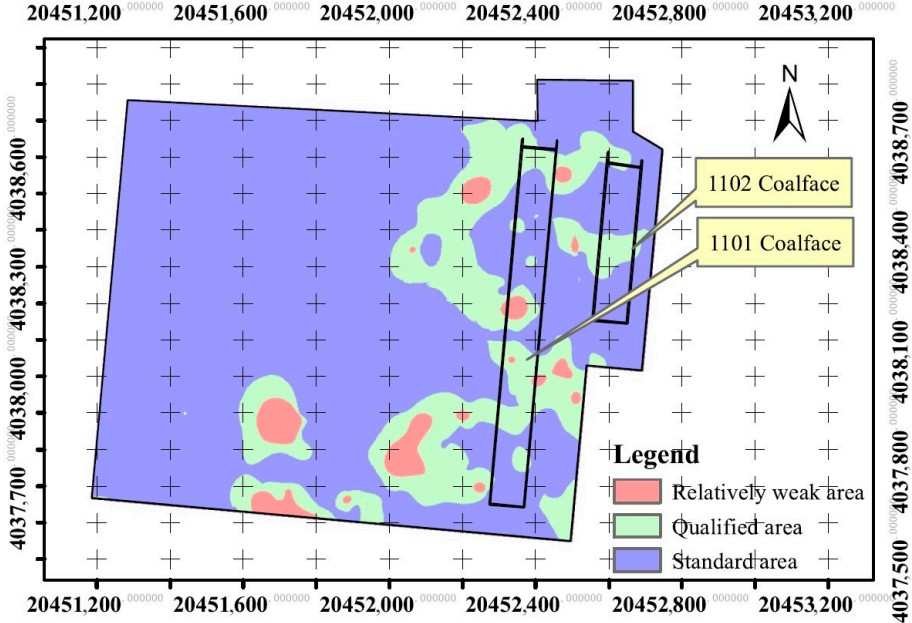

**Figure 5.** Evaluation diagram of grouting effect in the 1101 coalface floor under scheme II.

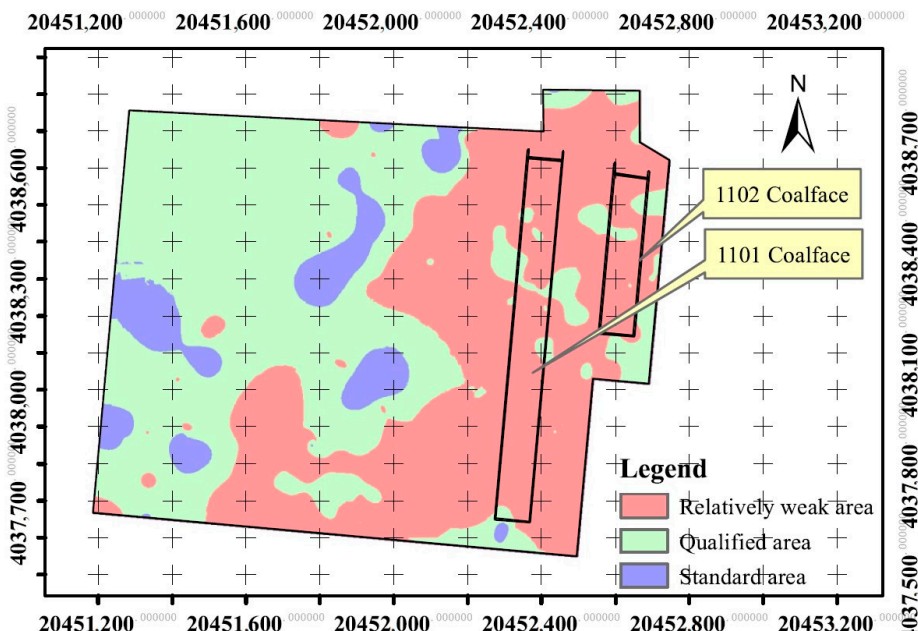

**Figure 6.** Evaluation diagram of grouting effect in the 1101 coalface floor under scheme III.

Scheme I: The reconstruction data of ground directional drilling and grouting based on the Xujiazhuang limestone aquifer in the floor of the 1101 coalface are shown in Table 1, and using ArcGIS for zoning evaluation, the evaluation results are shown in Figure 4.

Scheme II: The final grouting pressure is set to twice the water pressure, and the effect of grouting on the Xujiazhuang limestone aquifer is divided into three levels according to the grouting pressure: standard area (final pressure ≥ 10 MPa), qualified area (10 MPa ≤ final pressure < 8.5 MPa), and relatively weak area (final pressure < 8.5 MPa). Based on the directional drilling grouting transformation data regarding the Xujiazhuang limestone aquifer, ArcGIS is used for regional evaluation, and the evaluation results are shown in Figure 5.

Scheme III: The final grouting pressure is set to 2.5 times that of the water pressure, and the grouting effect of the Xujiazhuang limestone aquifer is divided into three levels according to the grouting pressure value: the standard area (final pressure ≥ 12.125 MPa), qualified area (12.125 MPa ≤ final pressure < 10.625 MPa) and relatively weak area (final pressure < 10.625 MPa). Based on the directional drilling grouting transformation data of the Xujiazhang limestone aquifer, ArcGIS is used for zoning evaluation, and the evaluation results are shown in Figure 6.

### 3.2. Verification of Mine Electrical Prospecting

#### 3.2.1. Geophysical Characteristics of the Study Area

Different rock formations have different conductivity values; generally, moving from clay to siltstone, fine sandstone, medium–coarse sandstone and limestone, the resistivity values gradually increase, and the resistivity value of limestone is the highest. Coal-bearing strata have a layered distribution characteristic, so the conductivity is relatively uniform in the lateral direction, and the electrical characteristics have consistency in the vertical direction. These characteristics are basically stable within the same region or exploration area, and are highly regular and comparable. When there are karst fractures, faults, collapse columns and other structures, they will have good conductivity due to the water-filled geological body or geological belt, and there will be obvious electrical differences from the surrounding rocks. The main lithology of the 1101 coalface strata is siltstone, sandstone, mudstone, coal seam, limestone, etc. The resistivity of mudstone is about 10–50 ohm-meters, and the resistivity of various sandstones is about 100–200 ohm-meters. The resistivity of limestone is as high as n × 100 or even n × 1000 ohm-meters when it is dense and complete.

The development of karst fractures, fractured zones, and water-filled parts greatly enhances the conductivity, and the resistivity can be reduced by one order of magnitude. This obvious electrical contrast constitutes a good geophysical basis for the application of the electrical method in mines to detect water-bearing structures and rich water-bearing regions.

### 3.2.2. Exploration Results

This project completed 90 electro-surveys in the 1101 coalface. After quality inspection, all technical indicators met the design and specification requirements. The detection results are basically consistent with the existing geological and drilling data, and can be used as the basis for geological hydrological analysis and underground water prevention designs. According to the detection results, eight anomalous low-resistance zones were identified within the detection range of the top and bottom plates of the 1101 coalface, named D1 to D8. The possibility of rich water presence in these anomaly zones is relatively high (Figure 7).

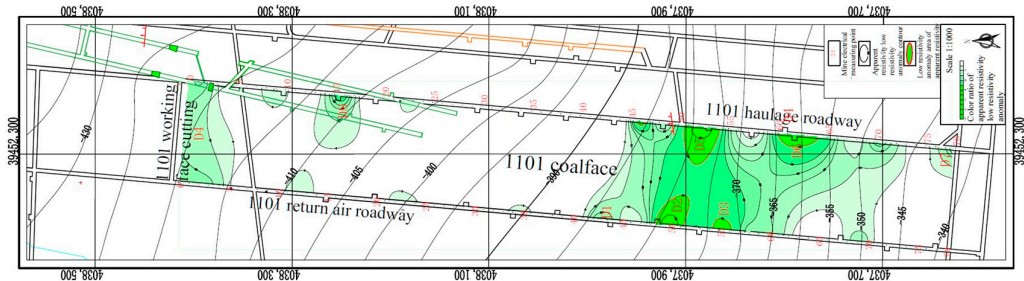

**Figure 7.** Isogram slice of the bedding, showing its resistivity and a low-resistance anomaly, of the Xujiazhuang limestone aquifer in the floor of the 1101 coalface.

### 3.3. Contrastive Analysis

The three schemes set out in the manuscript use different grouting pressure standards to test, and thus derive different results (as shown in Figures 4–6). We have sought to verify the accuracy of the three different results with measured data, such as that regarding the mine electrical method and drilling, so as to judge which scheme offers the optimal solution.

As shown in Figure 8, the results of the mine electrical method are very similar to the classification evaluation results shown in Figure 5. The D1 D6 anomaly points in the mine electrical method are similar to the central and southern anomaly areas of the 1101 coalface in Figure 5. The D4 and D8 anomaly points that appear in the mine electrical method are consistent with the location of the anomaly area at the northern cut of the 1101 coalface in Figure 5. The location of D7 in the mine electrical method is consistent with the abnormal area near the southern stop line of the 1101 coalface in Figure 5.

By comparing the ArcGIS evaluation test data with the mine electrical exploration data, it is found that the ArcGIS evaluation data of scheme II are well-matched with the data of mine electrical exploration, as shown in Figure 8. However, the ArcGIS evaluation data of schemes I and III show a significant discrepancy from the data of mine electrical exploration.

Therefore, under scheme II (setting the evaluation injection pressure to twice the water pressure), the effect of the Xujiazhuang limestone aquifer grouting can be divided into three levels according to the injection pressure values: standard area (final pressure ≥ 10 MPa), qualified area (10 MPa ≤ final pressure < 8.5 MPa), and relatively weak area (final pressure < 8.5 MPa). The results of zone evaluation using ArcGIS based on the directional drilling grouting data of the floor of the 1101 coalface are better. Since the accuracy of the mine electric method exploration results and ArcGIS evaluation data are both limited, the results of both can be integrated, and borehole verification can be conducted underground to improve the accuracy of the evaluation results.

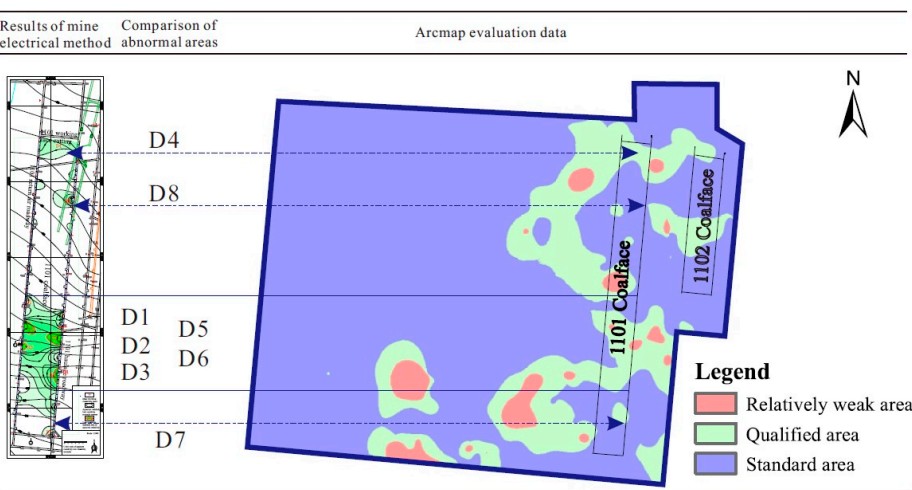

**Figure 8.** Comparative analysis of grouting reconstruction effect.

## 4. Discussion

### 4.1. Comparative Analysis of GIS Classification Evaluation and Underground Drilling Verification Evaluation

Firstly, a GIS classification evaluation based on grouting pressure can provide an overall risk evaluation of the pre-mining coal face in the form of a "face", with less workload, faster speed, strong operability, and relatively good accuracy. However, based on the verification and evaluation of underground drilling, a scientific drilling position design model is required, as the coverage area of single-hole detection is relatively small. If a coal mining face requires a large amount of drilling during construction, the pre-mining construction site will be limited, the workload will be large, the speed will be slow, and it will show a certain degree of operability, but the accuracy will be high.

Secondly, as shown in Figures 4–6, the GIS classification evaluation based on grouting pressure requires a large amount of grouting monitoring data, which therefore requires high accuracy. However, factors such as the depth of floor failure, the grouting quantity, the regional fault structure and the activation of hidden collapse columns are not considered, and there are still some hidden dangers. Therefore, GIS evaluation based on the grouting pressure used is suggested to further verify the area of the hidden danger via underground drilling before mining. Borehole verification and evaluation can be used to identify areas with lots of grouting and those with good geological structure.

Finally, drilling verification evaluation will damage the complete rock layer. If the borehole is not properly sealed, it can lead to great hidden dangers. Therefore, when evaluating the design of the drilling verification approach, it is necessary to consider the appropriate quantity, rather than assuming that more drilling is better.

Based on the above issues, the results of GIS classification evaluation based on grouting pressure can be appropriately combined with underground exploration drilling coordination verification evaluation, which can improve the accuracy of evaluation and reduce the workload of exploration drilling. Therefore, at present, the GIS classification evaluation method and the underground drilling verification evaluation method can be used together to improve the accuracy of grouting effect evaluation.

### 4.2. Optimization of Evaluation Method for Grouting Transformation Effect of Coalface

4.2.1. Coupling of Grouting Pressure and Grouting Volume

The amount of grouting reflects the development of water storage spaces, such as faults, fractures, and dissolution gaps, in the transformed rock layers. The amount of grouting cannot directly reflect the quality of the grouting effect. Therefore, GIS classification evaluation based on grouting pressure can be conducted first. Secondly, a GIS classification evaluation based on grouting volume will be conducted, and unqualified grouting areas

will be designated based on the contribution of two factors to the grouting effect. Finally, the combined results of the two classification methods can be used to further improve the accuracy of the evaluation.

4.2.2. GIS Evaluation and Collaboration with Geophysical Exploration and Dense Drilling

In order to improve the accuracy of the evaluation results, firstly, based on the GIS classification evaluation results of grouting pressure and grouting volume, geophysical verification is carried out using the mine electrical method, and the evaluation results of the two methods are combined. Secondly, underground drilling is performed for underground verification, and the results of the two methods are taken as their intersection. Finally, a scientific and accurate evaluation of grouting effect can be obtained. This method is feasible in theory, and it also represents the main focus of future research.

**5. Conclusions**

(1) Based on the data of grouting reinforcement in the floor of the 1101 coalface, an evaluation method using ArcGIS to determine the treatment effect of floor water hazards in a coal mining face was proposed;

(2) Following the comparison and analysis of the evaluation test data yielded by ArcGIS and exploration via the mine electric method, the results show that the water pressure threshold based on the GIS system is double the water pressure of the grouting reconstruction layer, thus representing the optimal solution;

(3) According to the comparative analysis between the data yielded by geophysical exploration and drilling and the model's experiment results, the feasibility and accuracy of grouting effect evaluation based on GIS using grouting pressure data are verified. The results of this research can provide a reference for the prevention and control of floor water hazards in deep coal seam mining in North China.

**Author Contributions:** Conceptualization, Y.H. and Q.X.; methodology, Y.H. and X.C.; software, J.L. and Y.H.; formal analysis, J.L. and Q.X.; investigation, Y.H. and Q.X.; data curation, J.L. and Y.H.; writing—original draft preparation, Y.H. and J.L.; writing—review and editing, X.C. and Y.H.; supervision, Y.H. and X.C. All authors have read and agreed to the published version of the manuscript.

**Funding:** This research was funded by the Natural Science Foundation of Jiangsu Province grant number BK 20221319.

**Institutional Review Board Statement:** Not applicable.

**Informed Consent Statement:** Not applicable.

**Data Availability Statement:** The data presented in this study are available upon reasonable request from the corresponding author.

**Acknowledgments:** This work was supported by the Natural Science Foundation of Jiangsu Province under Grant No. BK20221319. We are thankful for the scientific research project funds provided by the Geo-engineering Investigation Institute of Jiangsu Province.

**Conflicts of Interest:** The authors declare no conflict of interest.

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
