# Peer review of "Evaluation of Control Effect of Confined Water Hazard in Taiyuan Formation Coal Seam Mining in Huanghebei Coalfield"

_water, doi:10.3390/w15111973_

Round 1
Reviewer 1 Report
(1) According to the section “2.1. Overview of the research area” in the manuscript, Figure 1 is not suitable for characterizing the geological overview of the study area. More precisely, the scale of Figure 1 is too small, and Figure 1 should be more focused and small-scale.. Therefore, I think Figure 1 should be replaced by other Figures. The F Coal Mine in the North of the Huanghebei Coalfield should be clearly dislayed in Figure1.
(2) In Page 6: The grouting effect of Xujiazhuang Limestone aquifer is divided into three levels according to the grouting pressure value. This is very beneficial for research, but why do authors use the pressure range mentioned in the manuscript to divide it? Is it an industry standard or the authors' own standard? However, I see that different pressure standards were used in different scheme. Why is this? Please explain.
(3) In the section 4.1. Verification of mine electrical prospecting in the manuscript, the results of 4.1.1 can be presented in graphical form. Moreover, the content of 4.1.2 can be deleted. This is because the theory and principles of the application of electrical exploration in coal mines are very mature, not the innovation and content studied in the manuscript.
(4) Please state the innovation of this research. Is it just the application of ArcGIS system to the research on water inrush control by grouting? Even if this is the case, I think more quantitative results should be shown in the text, rather than simply describing engineering cases.
(5) In response to question 4, the abstract and conclusions need to be rewritten, adding some quantitative results to make the research more convincing. Moreover, the language of the manuscript also needs to be thoroughly polished and modified by native English speakers or professional organizations, and it is best to provide relevant certification.
(6) In manuscript, it was mentioned that "The mining of coal seams is affected by factors such as high ground stress, high water pressure, high temperature, and strong mining disturbance, among which the threat of high-pressure water hazard is particularly serious.". The follwing references should be added to support it. (1) Environmental Science and Pollution Research, 2022, 29(51), 77737–77754. https://doi.org/10.1007/s11356-022-21233-7. (2) Geomechanics and Geophysics for Geo-Energy and Geo-Resources, 2022, 8(2), 82. https://doi.org/10.1007/s40948-022-00396-0. (3) Environmental Science and Pollution Research, (2023). https://doi.org/10.1007/s11356-023-26279-9. They can be found by the input website address in browser.
Reviewer 2 Report
The authors present a methodology to evaluate injection grouting pressure by comparing grouting data to geophysical data on the example of a coal mine in Huanghebei Coalfield. The results and possible applications of the method should be of interest to control water hazards in coal mining, but in the current state of the manuscript this statement cannot be properly understood due to a number of scientific deficiencies. While the motivation for the study and the literature background are sufficiently described, the description of the methodology should be be comprehensively and thoroughly improved by detailed additions, a clear internal structure and, most important, by elimination of the indication that a methodological flaw could be present. Results should be presented in an own chapter. Figures can be improved with more information, more detailed and more precise captions as well as better integration into the text. Also, a discussion chapter comprising a critical confrontation with the method as well as grouting itself and possible fields of application of the method (incl. transferability, environmental risk assessment) should definitely be added together with appropriate references. Here it should also be emphasised what is special and new about the method presented and how it can extend existing methods, referencing to the topic of water hazards.

Round 2
Reviewer 1 Report
The manuscript can be accepted for publication now, and the quality of the manuscript has met my requirements.
Reviewer 2 Report
The authors have adressed a number of comments from the first review in parts, however, some comments have not been adressed at all resulting in the review not being adequately conducted. There are three main points still open: 1) The method of GIS classification is still not sufficiently described and it is not clear how an arbitrarily chosen classification of grouting pressure, which is the same in all three schemes, can contribute to an evaluation of the risk of water hazards. Figures 4 to 6 give the impression that weak zones are increasing, although this is only an effect of the classification. This needs to be described more clearly and the objective of the classification needs to be formulated. 2) Also, an improved internal structure and 3) a comprehensive discussion chapter are still missing.
Round 3
Reviewer 2 Report
Thank you for taking up my suggestions and modifying your manuscript! I think it has improved a lot to present your methodology to a broad readership in a reproducible and scientificly sound manner. It becomes clear now what the benefit of GIS evaluation can be, how it can be used and where further research is necessary. I now see great potential for the method to be used and further developed to ultimately help prevent serious accidents in underground mining. I wish you continued success in your research and further work.